# A Duality of Function: An Integrative Model of RACK1 as a Switch Between Translational and Signaling Hubs

**DOI:** 10.3390/ijms262311733

**Published:** 2025-12-04

**Authors:** Peter Kolosov, Nikita Biziaev, Elena Alkalaeva

**Affiliations:** 1Engelhardt Institute of Molecular Biology, The Russian Academy of Sciences, 119991 Moscow, Russia; bizyaev@eimb.ru (N.B.); alkalaeva@eimb.ru (E.A.); 2Institute of Higher Nervous Activity and Neurophysiology, The Russian Academy of Sciences, 117485 Moscow, Russia

**Keywords:** conventional PKCβII, RACK1, translation regulation, ribosomal proteins, synaptic plasticity, scaffold hubs

## Abstract

RACK1 (Receptor for Activated C Kinase 1) is a highly conserved scaffold protein that functions as a central integrator within diverse cellular signaling pathways. Initially identified as a receptor for activated Protein Kinase C, it is now recognized as a dynamic platform coordinating processes such as cell proliferation, migration, apoptosis, and immune responses. A defining feature of RACK1 is its ability to direct cellular fate by determining whether proteins are synthesized or degraded. However, a unified model explaining this functional pleiotropy has been lacking. In this review, we synthesize current knowledge to propose an integrative model centered on a functional dimorphism driven by RACK1’s localization and post-translational modifications. We posit that RACK1 operates in two primary, mutually exclusive states: a ribosome-associated monomer that supports the translation of specific mRNAs and quality control, and a free monomer or dimer that governs signaling cascades and gene expression. Phosphorylation at key sites, such as Thr50 and Ser146, acts as a molecular switch, spatiotemporally redistributing RACK1 between these pools. This mechanism allows the cell to rapidly reprogram its proteomic landscape in response to stimuli, pivoting between protein synthesis and stress adaptation. Our model resolves the apparent dichotomy of RACK1’s roles by framing it as a cellular “resource manager,” whose regulated switching between functional states ensures an optimal response to the extracellular environment, with significant implications for understanding cancer and neurodegenerative diseases.

## 1. Structure and General Properties of RACK1

### Structure of the Review

Given the pleiotropic nature of RACK1, this review is structured to first elucidate its universal molecular mechanisms—such as its role as a scaffold in core signaling pathways, ribosome-associated functions, and regulation by post-translational modifications. These sections focus on conserved principles across cell types. Subsequently, we highlight tissue- and cell-type-specific roles of RACK1, with a particular emphasis on the nervous system due to its dependence on localized translation and synaptic plasticity, while also referencing its critical functions in immune cells, cancer, and other contexts. This approach aims to provide a clear framework for understanding both the fundamental biology of RACK1 and its specialized adaptations in different physiological and pathological settings.

Receptor for Activated C Kinase 1 (RACK1) belongs to the WD40 protein family. Its structure comprises seven minidomains, known as WD repeats, which collectively form a β-propeller. This configuration allows RACK1 to interact simultaneously with multiple protein partners by binding them via distinct WD repeats [1,2,3,4]. The first identified binding partner of RACK1 was Protein Kinase C (PKC) [5,6].

RACK1 is highly conserved across evolution. Orthologs of human RACK1 have been identified in diverse unicellular and multicellular eukaryotes, including yeast, flowering plants, nematodes, and insects [1,4,7,8]. The amino acid sequence similarity between orthologs from evolutionarily distant species can be as high as 50%. However, the linker regions connecting the WD repeats are considerably less conserved and exhibit specificity across different orthologs [3,7,8,9,10]. These differences contribute to the functional variation in RACK1 among species. This review will focus primarily on mammalian RACK1; its plant orthologs are discussed in specialized reviews [7,11].

In mammals, RACK1 is ubiquitously expressed, although its expression levels vary significantly across different tissues such as the brain, liver, heart, and spleen [1,9,12] (The Human Protein Atlas: ENSG00000204628-RACK1 [13]). Notably, RACK1 expression undergoes dynamic changes in the nervous system during development, showing high levels in proliferating neuroblasts of the embryonic brain which subsequently decrease and become more region-specific in the adult brain, with prominent expression in structures like the cerebral cortex, hippocampus, and cerebellum [14].

Functionally, RACK1 is a canonical scaffold protein. It lacks intrinsic enzymatic activity and serves as a dynamic platform that concurrently recruits modifier and effector proteins, thereby modulating their localization, activity, and stability [1,2,3,15,16,17]. RACK1 is predicted to interact with hundreds of partners, with the functional consequences of these interactions having been characterized for dozens of them.

## 2. RACK1 in Intracellular Signaling

A primary function of RACK1 is to regulate key signaling enzymes, particularly kinases and phosphatases that control cellular processes through reversible phosphorylation. RACK1 interacts with a diverse set of these enzymes, including Protein Kinase C (PKC) isoforms, non-receptor tyrosine kinases (Src, Fyn), MAP kinases (JNK), AKT, and protein phosphatase PP2A, along with their respective substrates and regulators [1,15,18,19,20]. The functional outcome of these interactions is context-dependent. For instance, RACK1 stabilizes the catalytically active open conformation of the PKCβII and likely PKCε isoforms by acting as a scaffolding platform that binds and positions the activated kinase, thereby preventing its inactivation or premature degradation [1,15,18,19,20]. Conversely, RACK1 can also inhibit kinase activity. A key example is the Fyn kinase, which, when bound to RACK1, is unable to phosphorylate the glutamate receptor NMDAR (N-methyl-D-aspartate receptor). Activation of the cAMP/PKA pathway triggers the dissociation of this complex, leading to Fyn activation and subsequent receptor modification. This specific trimeric complex forms efficiently only in certain types of neuronal cells [1,3,20,21].

These neuronal-specific complexes exemplify RACK1’s ability to create specialized signaling contexts. However, RACK1’s role as a scaffold extends to many other cell types, where it similarly organizes critical signaling hubs. Beyond cytosolic enzymes, RACK1 forms specialized signaling hubs at the membrane by interacting with various transmembrane receptors. These membrane-associated complexes, which include GABA (GABAAR), acetylcholine (AChE-R), and glutamate (NMDAR) receptors alongside modifying kinases like PKCβII and Fyn, create unique molecular contexts essential for the specialized functions of cells in distinct brain regions, ultimately modulating organismal behavior [1,20,22,23]. The repertoire of RACK1’s membrane receptor partners extends beyond neurons. Its interaction with Gβγ subunits not only blocks signal propagation by Gβγ but also targets RACK1 to the plasma membrane, thereby facilitating cross-talk and balancing interactions between G-protein coupled receptors (GPCRs) and ligand-gated channels such as the NMDAR [1,3,20,24]. Furthermore, the RACK1 complex with the Androgen Receptor (AR) is required for AR phosphorylation by PKC or Src, which stimulates the receptor’s translocation to the nucleus [1,20,25,26].

RACK1 is also a crucial component of integrin receptor signaling and focal adhesion complexes, and participates in organizing the spectrin and actin cytoskeleton, as well as centrosomes. It forms various membrane-associated complexes with partners like the IGF-1 receptor, β-integrin, β-spectrin, and the kinases Aurora-A, AKT, FAK, and Src, as well as phosphatase PP2A. For example, the RACK1/β-spectrin interaction regulates the activity of the associated PKCβ [17]. The functional consequence of RACK1 binding is partner-specific: while the RACK1-Src complex inactivates the kinase, an external stimulus can trigger Src release to propagate signals from integrin receptors into the cytosol [27]. Intriguingly, within the context of the androgen receptor complex, Src appears to be activated and modifies surrounding target proteins [16,26,28]. RACK1’s partners can also bind with different forms of the protein. For instance, PP2A associates with the Tyr-302 phosphorylated form of RACK1, whereas β-integrin binds the unphosphorylated form. Switching between these states enables cells to shift between different strategic programs [1,3,29,30]. The complex of RACK1 with FAK and phosphodiesterase PDE4D5 is critical for cell polarization and migration [3,16,31]. Additionally, RACK1 recruits STAT proteins associated with JAK kinases and interferon signaling into its hubs [16,32]. Collectively, the signaling assemblies orchestrated by RACK1 are vital for cell adhesion and motility [1,3,20,33,34]. The precise regulation of these signaling hubs is critical for normal physiology, and their dysregulation is frequently implicated in pathological states such as cancer, where RACK1 often exhibits altered expression and function.

## 3. RACK1 in Cell Structure Organization and Fate Determination

RACK1 signaling hubs are critical for cytoskeletal organization and local protein synthesis. They play a key role in organizing the actin cytoskeleton [3,20,35,36] and are involved in pathways that regulate the local translation of β-actin—that is, the synthesis of β-actin directly in subcellular compartments. Ribosome-associated RACK1 binds the β-actin mRNA/ZBP1 complex and facilitates mRNA release for subsequent local translation, as shown in cellular and neuronal models [35,36]. This process involves Src kinase-mediated phosphorylation of RACK1 at Thr246, creating a binding site for ZBP1 (Zipcode-binding protein 1)-an RNA-binding protein that also serves as a Src substrate. ZBP1 is responsible for transporting β-actin mRNA and maintaining its translational repression [35,36]. Subsequent phosphorylation of ZBP1 by Src likely triggers the localized translation of the delivered mRNA in neuronal peripheries. Additional regulation of this translational activation comes from RNA-binding proteins associated with IGF signaling [37].

In cell division control, RACK1 contributes to centrosome function through its localization at centrosomes and spindle poles. It ensures proper centrosomal positioning of BRCA1, with their interaction being crucial for centriole number regulation [38]. Furthermore, RACK1 promotes centriole duplication by modulating PLK1 kinase activity during S-phase via its interaction with Aurora-A [38,39,40]. Notably, RACK1 overexpression can induce centriole amplification and overduplication in certain cell types [38,39]. The diverse composition and functions of RACK1-containing complexes often regulate competing cellular processes [1,3,29,30], highlighting the critical importance of molecular context in determining specific cellular responses [16,28].

In addition to these roles in structural organization and cell division, RACK1 is a pivotal regulator of cell fate. RACK1 significantly influences cell fate decisions by regulating proteins controlling apoptosis and stress responses. It recruits components of the ubiquitin-ligase system (such as Elongin-C) to mediate proteasomal degradation of specific targets [16,28,41]. This mechanism regulates both pro- and anti-apoptotic proteins like BimEL, thereby influencing cellular transformation potential. RACK1 also competes with HSP90 chaperone for HIF-1α (Hypoxia-inducible factor 1α) binding: while HSP90 stabilizes HIF-1α, RACK1 binding promotes its ubiquitination and degradation [1,28,42]. Under hypoxic conditions, degradation of the RNA demethylase FTO appears to promote apoptosis by displacing RACK1 from its complex with MTK1 kinase, thereby blocking MTK1-JNK1/2 signaling [43]. This regulatory loop helps restrict excessive hypoxic responses and angiogenesis [1,28,42,43].

The protein’s role in cellular stress extends to immune and inflammatory responses. RACK1 binds several interleukins and forms a hub essential for NLRP3 inflammasome activation, which coordinates innate immune responses, inflammation, and cell death [20,44]. Additionally, AMP-activated protein kinase (AMPK)-mediated phosphorylation of RACK1 at Thr50 stimulates assembly of the autophagy initiation complex (Atg14L-Beclin1-Vps34-Vps15) [20,45]. RACK1 can directly interact with core apoptotic regulators; for example, studies have shown that RACK1 promotes the dissociation of the Bax–Bcl-2 (or Bax–Bcl-XL) complex, stimulating Bax oligomerization and the progression of the mitochondrial apoptosis pathway [16,28,46]. Through complex formation with Casein Kinase II subunits, RACK1 stabilizes the kinase and activates NF-κB signaling, promoting G2/M cell cycle transition [47]. In immune cells, T-cell activation triggers rapid, concomitant redistribution of Lck (a Src-family kinase) and RACK1 to the forming immunological synapse [48]. RACK1 also participates in regulating mitophagy via the PINK1/Parkin pathway, particularly important in neuronal cells [49].

Beyond its cytoplasmic functions, RACK1 regulates nuclear gene transcription. It inhibits the transcriptional activity of p73α, thereby modulating expression of its target genes [28,50]. Through complex formation with 14-3-3ζ, RACK1 facilitates nuclear translocation and regulates chromatin remodeling and BDNF gene transcription [1,51,52]. The protein enhances transcription factor E2F1 stability by preventing its ubiquitination-a mechanism that can be disrupted in cancer cells [53]. Nuclear PKCα–RACK1–BMAL1 complexes contribute to circadian rhythm regulation [15,54], while RACK1 binding to Pax5 transcription factor stabilizes it and regulates B-cell development and function [55]. Finally, RACK1 interaction with viral E1A protein blocks its ability to modulate gene transcription and suppress growth [56].

Collectively, these findings indicate that RACK1 plays a key role in facilitating the selective translation of specific mRNAs. While these nuclear and transcriptional roles are important, RACK1’s predominant and evolutionarily conserved functions are executed in the cytoplasm, most notably through its stable association with the ribosome.

## 4. Ribosome-Associated Functions of RACK1 in Translation and Quality Control

A defining biochemical feature of RACK1 is its stable association with the 40S ribosomal subunit at the mRNA exit channel, mediated by multiple interactions with ribosomal proteins and rRNA [3,57,58,59]. It is believed that nearly every 40S subunit in the cell contains a tightly bound RACK1 molecule [3,60]. This ribosome-bound RACK1 serves as a platform to recruit various translation factors and regulatory proteins, thereby modulating translation [3,15,57]. A well-characterized example is the RACK1-mediated assembly of a complex between the 40S subunit and kinase PKCβII [3,61,62].

RACK1 regulates translation initiation indirectly through its interaction with eIF6. Because eIF6 binds the 60S subunit and prevents premature joining with the 40S subunit, RACK1-dependent modulation of eIF6 availability influences the efficiency of 40S–60S ribosomal subunit association [63,64]. As previously mentioned, PKCβII phosphorylates eIF6, triggering its release from the 60S subunit and enabling full ribosome assembly [63]. The ability of RACK1 to simultaneously bind both eIF6 and PKCβII suggests that the ribosome-tethered RACK1-PKCβII complex directly facilitates eIF6 phosphorylation [3,15,64]. The precise mechanism by which 40S-associated RACK1 regulates eIF6 on the 60S subunit remains unclear. One proposed model is a cascade mechanism wherein the RACK1-containing 40S subunit promotes eIF6 release from individual 60S particles. These newly liberated 60S subunits can then join other 40S particles, which in turn participate in further eIF6 release, enabling rapid and coordinated 80S ribosome formation [65].

Furthermore, PKC isoforms, bound with RACK1, can phosphorylate several other initiation factors, including the cap-binding protein eIF4E and subunits of the eIF3 complex (eIF3a and eIF3d), which is associated with the 40S subunit and essential for its function in translation initiation [66,67,68,69]. On the other hand, under conditions of eIF4E inhibition, RACK1 recruits eIF3d to the 43S pre-initiation complex, a process enhanced by active PKCβII. This stimulates eIF4E-independent translation, likely of a specific subset of mRNAs (e.g., those encoding chaperone HSP70) [69]. Additionally, it was shown that RACK1 is predominantly associated with polysomes and polyadenylated mRNAs, and the translation efficiency of these mRNAs depends on PKCβII activation [70].

As noted earlier, RACK1 can shape the cellular translatome by influencing the selective and spatially organized translation of specific mRNAs. For instance, it regulates local translation by modulating translational repressors like ZBP1 and FMRP in specific cellular compartments [35,36,71]. The protein LARP4, which protects mRNAs from deadenylation and degradation, interacts with ribosome-bound RACK1 and stimulates the translation of mRNAs containing AU-rich elements (ARE) [72]. Additionally, RACK1 enhances the internal ribosome entry site (IRES)-dependent translation of several viral mRNAs (e.g., poliovirus (PV), hepatitis C virus (HCV), and cricket paralysis virus (CrPV)) [73,74,75]. Notably, RACK1 knockdown suppresses viral IRES-dependent translation more strongly than cellular cap-dependent translation [75]. Collectively, these findings indicate that RACK1 plays a key role in the selective translation of specific mRNAs. This is likely achieved by forming specialized regulatory platforms on the 40S subunit and is influenced by mRNA features such as length, secondary structure, and the presence of specific cis-regulatory elements [65,67,71,76,77,78].

RACK1 is also involved in quality control mechanisms for both newly synthesized proteins and mRNAs. One of its partners on the ribosome is the kinase JNK. Under cellular stress, JNK is activated by PKC and phosphorylates the elongation factor eEF1A2 (isoform 2), which triggers the proteasomal degradation of the nascent polypeptide chain. The activation of JNK itself also depends on RACK1. It is proposed that under stress conditions, which increase the likelihood of defective protein synthesis, RACK1 directs JNK to polysomes to initiate this quality control pathway [3,15,79].

Another quality control mechanism involving RACK1 is related to mRNA degradation. In cooperation with the ubiquitin ligase ZNF598, ribosome-bound RACK1 promotes ribosome-associated quality control (RQC) on ribosomes stalled at poly(A) sequences. The RACK1-containing complex stabilizes the interface of a stalled disome (a ribosome dimer), leading to regulatory ubiquitination of specific 40S ribosomal proteins (RPS2, RPS3, RPS20) [80,81].

Moreover, RACK1 interacts with components of the microRNA-mediated mRNA decay pathway, such as Ago2 and KSRP (within the RISC complex), recruiting them to the translating mRNA [82]. This suggests a role for RACK1 in coupling mRNA translation with its degradation by facilitating the interaction between the RISC complex, the mRNA, and the translation machinery [1,28,83]. Furthermore, some viruses can exploit RACK1 to manipulate the efficiency of the host cell’s protein degradation system for foreign proteins [73].

## 5. RACK1 as a Cellular Integrator: Implications in Physiology and Disease

At the cellular level, RACK1 is essential for numerous fundamental processes, including cell differentiation, polarity establishment, adhesion, migration, division, immune responses, stress adaptation, and cell death. Its functions are particularly critical in neural and immune cells [1,15,16,17,18,20,28,41,84,85,86,87,88,89,90,91]. A notable example is its decisive role in regulating synaptic plasticity, where it mediates the induction of long-term depression (LTD) without altering the expression of postsynaptic proteins [92].

The biological importance of RACK1 is firmly established by knockout phenotypes. Mice with impaired RACK1 synthesis exhibit defects in cell migration and translation [15], while complete gene knockout results in embryonic lethality [93]. One of the most striking morphological consequences of RACK1 deficiency is the disruption of melanocyte biogenesis, a process fundamentally dependent on cell migration [15,93]. Interestingly, this phenotype mirrors that observed in natural mutants of the ribosomal protein rpL24, indicating that RACK1’s influence extends beyond migration to encompass general protein synthesis pathways [94].

The functional versatility of RACK1 extends to host–pathogen interactions, as various viruses and intracellular pathogens exploit RACK1 at different stages of their life cycles. Dysregulation of RACK1 is also frequently associated with pathological conditions including immunodeficiencies, cancerous transformation, and neurodegenerative disorders such as Alzheimer’s and Parkinson’s disease [1,15,16,17,18,20,28,41,73,84,88,91,95,96,97]. The pathological outcomes of RACK1 dysfunction vary significantly across different cell types, likely reflecting the specialized molecular context of RACK1-mediated hubs in each cellular environment. This cell-type specificity presents considerable challenges for developing targeted therapies and diagnostic strategies for RACK1-associated pathologies.

For comprehensive coverage of specific RACK1 functions, we direct readers to specialized reviews cited throughout this article, covering its roles in: signaling and cellular dynamics [1,15,28,65]; centrosome organization [38]; ribosome association [3]; stress granules and neuronal translation [95]; cancer [16]; nervous system development [84]; Alzheimer’s disease [20]; and viral infection [73].

The remarkable functional diversity of RACK1 and its presence in multiple cellular compartments raise a fundamental question: how is such pleiotropy achieved molecularly? This versatility likely stems from the intrinsic structural plasticity of RACK1, which enables it to function as a universal adapter protein. By interacting with different partners in specific spatiotemporal contexts, RACK1 generates a wide spectrum of biological effects. In the following section, we will examine the mechanisms underlying this dynamic behavior and propose a unified model of RACK1 functioning within the cell.

## 6. Post-Translational Modifications of RACK1

The RACK1 protein undergoes an extensive array of post-translational modifications that precisely regulate its diverse cellular functions. Multiple phosphorylation sites have been identified at residues Thr50, Tyr52, Ser110, Ser146, Tyr228, Tyr246, Ser276, Thr277, Ser278, Ser279, and Tyr302 [1,3,8,10,20,98,99,100,101].

### 6.1. Phosphorylation-Mediated Regulation

AMP-activated protein kinases (AMPK) phosphorylate RACK1 at Thr50, promoting its association with autophagosomal components VPS15, ATG14L, and Beclin 1 to stimulate autophagosome assembly [20,45,99]. This modification also triggers RACK1 dissociation from ribosomes, thereby inhibiting its translational functions while activating autophagy. Notably, wild-type RACK1 overexpression reduces infarct size, neuronal death, tissue loss, and neurobehavioral dysfunction, whereas the phospho-deficient T50A mutant produces opposite effects [99]. Paradoxically, Thr50 phosphorylation can also enhance viral infection by facilitating RACK1 interaction with interferon transcriptional factor IRF3, suppressing IRF3 phosphorylation and subsequent interferon synthesis [98].

Phosphorylation at Tyr52 by c-Abl kinase enhances RACK1 interaction with FAK, influencing cell adhesion and migration [1,102]. During fulminant hepatitis, Ser110 phosphorylation increases RACK1 binding to ubiquitin-conjugating enzyme E2T, promoting its own ubiquitination and degradation [100].

Ser146 phosphorylation induces RACK1 dimerization, which is essential for recruiting the Elongin-C ubiquitin ligase complex and subsequent HIF-1α degradation. Dephosphorylation of this residue by calcineurin phosphatase prevents dimerization and stabilizes HIF-1α [1,3,103]. Structural analysis suggests casein kinase 2 (CK2) mediates Ser146 phosphorylation [103], consistent with RACK1’s known ability to form complexes with CK2 subunits [47].

Src kinase phosphorylates Tyr228 and Tyr246 within its RACK1 binding site [3,101]. Interestingly, vaccinia virus kinase phosphorylates Ser276, Thr277, Ser278, and Ser279—residues not modified in uninfected cells. These phosphorylated residues cluster within an extended loop region, and their modification enables selective translation of viral mRNAs and adenosine-repeat-containing mRNAs. Surprisingly, this viral-induced phosphorylation mimics the negatively charged region found in plant RACK1 orthologs [8,10]. Indeed, introducing a negative charge at Ser278 via phosphomimetic mutation (S278D) distorts 80S ribosome conformation similarly to certain IRES elements, sustaining cap-independent translation [104].

Tyr302 phosphorylation increases RACK1 affinity for phosphatase PP2A without affecting β-integrin binding, shifting the balance between these competing interactions and potentially reducing cell migration efficiency [1,3,29].

Based on in silico modeling, a hypothesis has been proposed that additional phosphorylation of Ser63, Thr86, Ser276, Thr277, Ser278, and Ser279 may also contribute to the release of RACK1 from the ribosome. The corresponding potential kinases predicted for these modifications are AMPK1/2, ULK1/2, or PKR, with Jnk/SAPK1 and GCN2 being less likely candidates [105].

### 6.2. Additional Modification Types

Beyond phosphorylation, RACK1 undergoes poly(ADP-ribosyl)ation (PARylation), which further stimulates its phosphorylation and dimerization, ultimately promoting HIF-1α degradation [106]. Lys130 acylation prevents RACK1 ubiquitination at Lys6, Lys33, and Lys48, thereby enhancing its cellular stability [107]. Additionally, some cancer cells exhibit reduced expression of the E3 ligase responsible for RACK1 ubiquitination, correlating with elevated RACK1 protein levels [107].

Mono(ADP-ribosyl)ation (MARylation) facilitates RACK1 translocation to stress granules [106]. While initial reports suggested O-GlcNAcylation occurred during arsenite-induced stress granule formation [1,108], subsequent research identified Ser122 as the modification site and demonstrated that this modification actually enhances protein stability, ribosome binding, and PKCβII interaction, ultimately increasing eIF4E phosphorylation and translation efficiency, particularly in cancer cells [109].

In summary, RACK1 is subject to numerous post-translational modifications that dynamically regulate its function. However, for many modifications, the corresponding enzymes, including specific kinases and phosphatases, remain unidentified, and their structural impacts on RACK1-organized functional hubs are not fully understood.

## 7. RACK1 Homodimerization

An important regulatory mechanism underlying RACK1’s functional diversity is its ability to form homodimers and higher-order oligomeric complexes. The key event triggering RACK1 dimerization is phosphorylation at Ser146 [103,106]. Structural data indicate that dimerization involves the WD4 repeat [110], consistent with predictions of the dimerization interface based on structural analysis [4].

Experimental evidence confirms the existence of RACK1 dimers. For instance, two RACK1 molecules can simultaneously bind both Fyn kinase and the NR2B subunit of the NMDA receptor—an interaction impossible in the monomeric state [21,110]. The critical role of the dimeric form is particularly evident during the simultaneous recruitment of Elongin-C and its target protein HIF-1α, as their binding sites overlap in monomeric RACK1 [103,106]. Additional evidence suggests that the equilibrium between dimeric and monomeric RACK1 regulates Ras kinase activity, and that dimerization may be associated with RACK1 phosphorylation by Src kinase, although the specific phosphorylation site remains unidentified [111].

Under basal conditions, RACK1 exists predominantly in monomeric form. In vitro data show that non-phosphorylated RACK1 is approximately 90% monomeric, with only about 6% forming dimers and roughly 2% each forming tetramers and higher oligomers [112]. The dimer dissociation constant (~50 μM) [112] is two orders of magnitude higher than physiological RACK1 concentrations in HEK293T and HeLa cells (Biziaev N.S., personal communication), making spontaneous cytosolic dimerization extremely unlikely. Therefore, the transition to the dimeric state represents a regulated process triggered by specific phosphorylation events and potentially other modifications, rather than occurring spontaneously.

Thus, RACK1 oligomerization may function as a molecular switch that alters its spectrum of preferred protein partners and likely its subcellular localization. It can be hypothesized that the dimeric form serves as a platform for modifier enzyme/target protein interactions in gene expression regulation and stress signaling, while the monomeric form primarily associates with ribosomes and participates in translational regulation.

## 8. Stability, Dynamics, and Subcellular Localization of RACK1 Complexes

The functional regulation of RACK1 extends to its dynamic partitioning among different protein partners and the consequent localization of the resulting complexes within the cell. RACK1-containing complexes have been identified in diverse locations including the nucleus, plasma membrane, stress granules, ribosomes, cytosol, proteasomes, and inflammasomes [1,2,3,15,16,17,20,29,30,44,57,58,59,95]. This dynamic distribution is regulated in part by post-translational modifications, such as Thr50 phosphorylation [99], MARylation [113], and Ser146 phosphorylation-induced dimerization [103,106]. However, modifications alone cannot account for the full diversity of RACK1 hubs. The existing composition of formed complexes substantially influences subsequent partner binding and ultimate hub localization.

A striking example of this composition-dependent regulation is RACK1’s dual role in the Hedgehog pathway. In the absence of ligand (Hh), RACK1 promotes formation of the Ci–RACK1–Cos2 complex, leading to Ci proteolysis. Upon Hh binding, RACK1 dissociates from this complex and instead forms a trimeric complex with Smo and Usp8, resulting in Smo deubiquitination and subsequent stabilization/accumulation at the membrane [114]. Another illustration comes from Alzheimer’s disease pathology, where Aβ oligomers reduce RACK1 distribution in the membrane fraction of cortical neurons. This Aβ-mediated decrease in membrane association may underlie observed impairments in muscarinic regulation of PKC and GABAergic transmission [115].

The subcellular localization of PKC–RACK1 complexes is primarily governed by isoform-specific features of PKC. For instance, the PKCα–RACK1–BMAL1 complex translocates to the nucleus to influence circadian rhythm regulation [1,15,54], while PKCβII has been detected in mitochondria [116]. In these contexts, RACK1 likely serves as a stabilizing scaffold for the activated kinase rather than being the primary determinant of its localization. A hypothesis has been proposed that cytoplasmic PKC may also act as a molecular ‘sink’ for a pool of RACK1 [1,117], although the dynamics of this interaction require further investigation.

A more substantiated cytosolic anchoring mechanism for RACK1 involves ribosomes. As RACK1 is tightly associated with the 40S [3,60], it can mediate the delivery of ribosomes, specific mRNAs (such as actin mRNA), and kinases to particular cellular compartments (e.g., dendrites), thereby participating in local translation [3,15,17,65,84]. Indeed, for example, in spreading and migrating cells, ribosomes and translation factors accumulate at adhesion complexes enriched with integrins [15,65,118].

Within the cytosol, RACK1 distribution is heterogeneous, particularly evident in neurons where it localizes to cell bodies and dendrites but is absent from axons [1,14,70]. Its presence in the neuropil correlates with the translational activity of these zones, and when translation is disrupted, RACK1 saturation in these areas significantly decreases alongside other ribosomal proteins [119]. This indicates that dendritic RACK1 exists predominantly in a ribosome-associated state. Furthermore, the composition of ribosomal proteins is dynamic, and RACK1 is among those actively exchanging between different ribosomal particles [119], suggesting high mobility and function as a shuttle for regulatory complexes between ribosomes. Supporting this close ribosome association, RACK1’s expression profile in human cells correlates most strongly with genes encoding small ribosomal subunit proteins (The Human Protein Atlas: ENSG00000204628-RACK1 [13]). Collectively, these data position RACK1 as a physical link between the ribosome and certain signaling mechanisms, enabling translation modulation in response to diverse cellular stimuli.

Another crucial factor is the short half-life of free RACK1 in the cytosol (only a few hours), underscoring the importance of rapid and efficient complex formation for its stabilization [107,109,120,121]. Uncomplexed RACK1 is subject to regulated ubiquitination and degradation [120,122]. Overexpression leading to accumulation of free RACK1 inhibits cell cycle progression [121] and autophagy [105]. This feature is used by viruses which can disrupt cellular signaling and defense systems inducing RACK1 overexpression [73]. Notably, overexpression of the R36D/K38E mutant, which is ribosome-binding deficient [80], reduces its half-life and accelerates degradation [109].

These findings collectively portray RACK1 as a platform that directs signaling and effector complexes to specific cellular locations in response to stimuli [1,15,16]. However, the detailed mechanisms governing RACK1’s targeted delivery and its overall dynamics remain incompletely understood. Furthermore, varying expression levels of RACK1 and its partners across cell types likely shifts the balance between different complexes and their localization, explaining why RACK1 exerts non-identical effects in, for example, different brain regions [1]. Disruption of this delicate localization balance is proposed to readily lead to cellular physiological failures and contribute to pathology, particularly in the nervous system [1,17]. Gaps in understanding this balance hinder development of compounds that can selectively target specific RACK1 functions in pathological states [67,95]. Intriguingly, antibody library screening has identified an anti-RACK1 antibody that effectively suppresses cancer cell proliferation, though its precise molecular mechanism and which specific RACK1 hub it disrupts remain unestablished [123]. This highlights the potential utility of developing an integrated model of RACK1 function to enable targeted rather than random therapeutic discovery.

Finally, a fundamental question remains: how are RACK1’s signaling, transcriptional, proteasomal, and ribosomal hubs interconnected? Do separate physical complexes exist between which the RACK1 pool is balanced, or can RACK1 form unified “megahubs” that coordinate these processes directly on translating ribosomes? Structural capabilities allow RACK1 to simultaneously bind the ribosome and several kinases, in principle permitting coupling of signaling and translation [3,15,17,28,65]. Indeed, ribosomes interact with numerous effectors, and their composition (the ribosome proteome) is heterogeneous [124]. The critical importance of the ribosome-associated form of RACK1 is underscored by findings that neuronal expression of the ribosome-binding-deficient R36D/K38E mutant [80] causes cellular defects resembling those upon RACK1 knockdown [71,125]. Expression of this mutant also reduces binding of various RNA-binding proteins (e.g., α-Scp160, TDP-43, FMRP), kinases (PKCβII), and translation factor eIF3d to the translation apparatus [67,69,71,126]. As noted, binding of certain partners to RACK1 is mutually exclusive [21,103,106,110]. Therefore, while RACK1 can form large hubs and theoretically unite several functional modules, the formation of a “megahub” simultaneously accommodating all its hundreds of partners seems improbable. This indicates fundamental heterogeneity among the formed hubs.

## 9. Ribosome-Associated vs. Free RACK1: Mutually Exclusive States Enabling a Switch Between Translation and Cellular Regulation

Given that the ribosomal hub of RACK1 is the most thoroughly characterized, our model focuses on its core functional dichotomy: the ribosome-associated state versus all free forms. As data on other RACK1 hubs (e.g., in stress granules, inflammasomes) accumulates, this simplified dichotomy can be expanded into a network model where nodes represent specific functional states of RACK1 and edges represent regulated transitions between them. Currently, constructing such a comprehensive network is impeded by a lack of structural data on the precise interactions within each of these complexes.

The choice of the ribosome-associated state as the central node for our current model is grounded in robust experimental evidence. Analysis of polysome profiles, a method that separates ribosomal complexes by mass in a sucrose gradient, unequivocally shows that the vast majority of cellular RACK1 is detected in heavy polysomal fractions, while the ribosome-free fraction is barely detectable [8,10,74,104]. This indicates that the primary pool of RACK1 is constitutively and stably associated with ribosomes. Consequently, the ribosomal hub is not merely one of many possible states but the dominant, or “home,” state of RACK1, making it the logical reference point for any functional model.

Ribosome-bound RACK1 is located on the solvent-exposed surface of the 40S subunit. Although a significant portion of its surface area contacts the ribosome in the human complex [127], the remainder is available for other interactions (Figure 1A). Structural analysis by Nielsen et al. [3], initially based on the yeast ribosome but expected to be conserved in mammals [127], identified potential partner-binding sites on ribosome-associated RACK1. According to this analysis, surfaces of the WD3, WD5, WD6, and WD7 repeats remain largely accessible, while WD1 is almost entirely, and WD2 and WD4 are partially, occluded by the ribosome (Figure 1B).

Thus, ribosome-associated RACK1 can potentially interact with: PKC kinases via accessible regions of WD2 and WD3; its substrates, the initiation factors eIF3a (which lies on the 40S surface and may interact with RACK1 via parts of WD1, WD2, and WD3) and eIF4E (via WD6 and/or WD7); β-integrin via free regions of WD4 and WD5; and Src kinase and the protein ZBP1 (which modulates localized β-actin translation) via a region of WD6 near Tyr246 [3] (Figure 1B). This architecture enables physical coupling of the ribosome to signaling complexes in focal adhesions and the cytoskeleton. Conversely, binding sites for partners such as Fyn kinase and the NR2B subunit of the NMDA receptor (located in the WD1 repeat), along with a significant portion of the FAK binding site (spanning the end of WD1 to the start of WD2), are occluded by the ribosome, making their interaction with the ribosome-bound form unlikely (Figure 1B) [3,21,110].

A key example of a functionally significant ribosome-free RACK1 hub is the complex regulating the stability of HIF-1α, the central transcriptional factor in the hypoxic response. While HIF-1α degradation under normoxia is mediated by the VHL complex, an alternative, VHL-independent pathway exists where RACK1 plays a critical role. RACK1 competes with the HSP90 chaperone for binding to HIF-1α. To initiate degradation, RACK1 must simultaneously recruit the Elongin-C ubiquitin ligase complex. However, the binding sites for HIF-1α and Elongin-C on RACK1 overlap, making their simultaneous binding possible only in the RACK1 dimer (Figure 1B) [42,103]. The binding site for HIF-1α and Elongin-C is located in the WD6 repeat [42,103], which overlaps with the eIF4E binding site [3], and thus is unlikely to be efficiently engaged while RACK1 is ribosome-bound (Figure 1B). Furthermore, the dimerization site involving Ser146 (which is also the interaction site for calcineurin) is located in the WD4 region, which interacts closely with the ribosome, casting doubt on its accessibility in the ribosome-bound state. This is corroborated by the fact that no structural study of ribosome-bound RACK1 has observed a bound dimer [3].

Therefore, dimerization triggered by Ser146 phosphorylation acts as a molecular switch activating this degradation pathway. The reverse process-dephosphorylation of Ser146 by calcineurin (a calcium/calmodulin-dependent phosphatase, PP2B, that binds RACK1)-disrupts the dimer. In the absence of dimerization, Elongin-C cannot be recruited, and HIF-1α is stabilized [42,103]. The functional significance of dimerization is confirmed by the finding that expression of the S146A mutant, incapable of dimerization, blocks HIF-1α degradation [42,103].

A similar spatial conflict exists for other nuclear functions of RACK1. For instance, the transcription factor GATA4 interacts with RACK1 via WD4 and WD5 repeats [128], sites also involved in ribosome and β-integrin binding [3]. Given the predominantly cytoplasmic localization of ribosomes, it follows that RACK1’s functions related to transcriptional regulation in the nucleus are carried out by its free, not ribosome-associated, form.

Based on the evidence, we propose that RACK1 functions through two primary, mutually exclusive states:

**The Ribosome-Associated State (Monomer):** In this predominant form, RACK1 serves as a platform for translation regulation by recruiting kinases (PKCβII, Src), phosphatases (PP2A), and RNA-binding proteins (ZBP1) to modulate factor phosphorylation and mRNA selection. This state is particularly crucial for translating mRNAs with IRESs, microRNA-regulated transcripts, and stress-sensitive mRNAs, while also participating in ribosome-associated quality control. Rather than being merely structural, ribosome-bound RACK1 functions as a signaling regulator that connects external stimuli to translational control and potentially couples translation with membrane signaling.

**The Free State (Monomer or Dimer):** This form engages in signaling pathways and regulates gene expression at transcriptional and post-translational levels. Transition to this state, triggered by events such as Thr50 phosphorylation by AMPK, induces RACK1 dissociation from ribosomes, suppresses translational functions, and activates new roles including autophagosome assembly or dimerization for HIF-1α degradation. The connection between calcium signaling, calcineurin, and RACK1 further suggests this platform integrates ionic and metabolic signals into transcriptional regulation. This functional switching typically occurs during stress, where RACK1 departure from ribosomes halts translation and activates stress responses like autophagy.

We further hypothesize that more subtle switching mechanisms might operate without complete ribosomal dissociation. Phosphorylation of residues in the conformationally plastic “knobb” loop (Ser276, Thr277, Ser278, Ser279) could act allosterically, inducing conformational changes that make the Ser146 dimerization site accessible while maintaining ribosomal association. This would enable local functional switching between translation and protein stability control, though this hypothesis requires experimental validation.

## 10. Integrative Model of RACK1 Function

Synthesizing these findings, we propose an integrative model wherein molecular switches in RACK1 determine its functional state and consequent cellular responses (Figure 2). The model centers on RACK1’s capacity to integrate diverse signals as a central processing unit that dynamically connects phosphorylation/dephosphorylation enzymes, translation factors, and protein degradation components. The balance between RACK1’s functional states is governed by post-translational modifications that direct cellular resources along specific pathways.

The model’s cornerstone is RACK1’s functional dimorphism based on localization. The ribosome-associated form maintains protein synthesis by controlling specific mRNA translation and polypeptide quality, while the free form (monomer or dimer) governs signaling cascades, stress responses, and gene expression regulation. Phosphorylation at key sites (Thr50, Ser146) thus serves as molecular switches enabling spatiotemporal redistribution of RACK1 from protein synthesis to degradation and stress adaptation programs. Biologically, this allows RACK1 to programmatically define the cellular proteomic landscape in response to external stimuli, determining which proteins to synthesize versus degrade.

The model describes several key switching scenarios that reveal the causal link from a molecular event to a cellular outcome (Figure 2):

**Basal State:** Ribosome-Associated Monomer. This is the “home” state of RACK1, where it maintains baseline and stimulated local translation, controls protein quality, and couples translation with membrane signaling. The free, unphosphorylated monomer is virtually undetectable in the cytosol due to its instability and high affinity for ribosomes.

**Switching to Stress Response Mode:** AMPK-mediated phosphorylation of Thr50 triggers RACK1 dissociation from the ribosome. Given the inherent instability of the free monomer, this switch acts as a rapid “trigger,” halting resource investment in protein synthesis and redirecting them toward stress adaptation and survival via processes like autophagy.

**Switching to Protein Stability Control:** In the free state, CK2-mediated phosphorylation of Ser146 induces RACK1 dimerization, facilitating the assembly of a complex with Elongin-C for targeted degradation of substrates such as HIF-1α. This provides a mechanism for suppressing angiogenesis under normoxia or fine-tuning the hypoxic transcriptional response.

**System Reset:** Calcium-activated calcineurin dephosphorylates Ser146, dissolving the dimer and promoting RACK1’s reassociation with the ribosome. This reset mechanism ensures a return to the basal translation program upon cessation of stress, the dynamic and reversible nature of the regulatory cycle.

A critical, testable prediction of this model is the structural incompatibility between the dimeric form and the ribosome-associated state. Experimental verification is needed to determine if a ribosome-bound dimer is possible and, if so, whether the individual monomers within such a complex could be differentially modified, for instance, if the distal monomer could be phosphorylated at Thr50, and what functional implications this might have (Figure 2).

By identifying these key regulatory nodes, the model provides a framework for understanding pathogenesis. Malfunctions in these molecular switches (e.g., aberrant phosphorylation) represent a plausible mechanism underlying diseases associated with disrupted proteostasis, signaling, and translation, such as neurodegeneration and cancer. Future work, particularly the quantitative assessment of fluxes between RACK1 states in living cells, will be crucial for fully elucidating its integrative role in cellular physiology [15].

## 11. RACK1 in Neurons: Local Translation and Synaptic Plasticity

Neurons provide an ideal system for demonstrating the predictive power of the integrative RACK1 model. Their high polarization, dependence on local protein synthesis in dendrites, and the critical need for precise control of synaptic transmission make RACK1-mediated mechanisms essential for neuronal function [20,84,95].

RACK1 is present in neuronal cell bodies and dendrites, including dendritic spines, and its tight association with ribosomes indicates a central role in regulating local protein synthesis [36,84]. A key aspect is its function as a ribosomal scaffold/adaptor: it recruits signaling kinases and RNA-binding proteins (RBPs, e.g., ZBP1) to the ribosome and thus mediates context-dependent regulation of specific mRNA translation in response to synaptic signals. For example, RACK1 mediates Src-dependent phosphorylation and subsequent derepression of β-actin mRNA translation via ZBP1, which is necessary for actin cytoskeleton remodeling in growth cones and dendrites [35,36]. Thus, consistent with the model, ribosome-associated RACK1 in neurons maintains both basal and activity-induced synthesis of proteins crucial for synaptic plasticity, such as Arc and CaMKIIα.

Experimental evidence shows that synaptic activity, such as NMDA receptor stimulation, alters RACK1 phosphorylation and subcellular distribution [21,76,110,129,130]. We interpret the accumulated data to suggest that RACK1 can function not merely as a static ribosomal component, but as a dynamic platform whose post-translational modification (e.g., phosphorylation) and partner swapping may translate receptor activation into changes in the synaptic translational landscape. It is conceivable that depending on the stimulation context, RACK1 phosphorylation could either locally restrict or activate protein synthesis. According to our model, the transition of RACK1 to its free state (e.g., through Thr50 phosphorylation) in response to metabolic stress or intense stimulation would lead to local translational suppression and potential activation of degradation pathways, thereby serving as a mechanism for synaptic homeostasis that prevents hyperexcitability.

The model posits the following functional scenario for RACK1 in neuron:

**Resting Neuron:** RACK1 primarily resides in a ribosome-associated state in dendrites, maintaining basal translation levels necessary for synaptic structure and response readiness.

**Moderately Activated Neuron:** Local calcium influx through NMDA receptors activates calcineurin. RACK1 dephosphorylation by calcineurin promotes its monomeric, ribosome-bound form, potentially enhancing local translation required for long-term potentiation (LTP) and synapse growth.

**Neuron Under Metabolic Stress or Hypoxia:** AMPK activation induces RACK1 phosphorylation at Thr50, causing ribosomal dissociation and transition to the free state. This switching globally suppresses energy-intensive dendritic translation while potentially activating autophagy and protein degradation pathways (e.g., through dimerization), promoting cellular survival under adverse conditions.

**Neuron with Imbalanced Signaling:** Persistent shift toward free, dimeric RACK1 forms incapable of supporting translation may underlie impaired synaptic plasticity in neurodegenerative diseases such as Alzheimer’s, where disrupted membrane association and function of RACK1 have been observed [20,115].

In summary, RACK1 acts as a key integrator in neurons, connecting synaptic, calcium, and metabolic signals with the local protein synthesis machinery. The integrative model finds clear illustration here: transitions between ribosome-associated and free RACK1 states contribute to determining synaptic responses by influencing whether synapses strengthen through new protein synthesis or modulate their activity toward suppression to maintain homeostasis. It should be emphasized that RACK1 represents just one of multiple plasticity mechanisms, and its specific contributions require further quantitative investigation within the broader regulatory network. Nevertheless, studying these transitions in living neurons remains crucial for understanding the precise mechanisms underlying learning, memory, and the pathogenesis of neurological disorders.

## 12. Current Challenges and Future Perspectives

Our analysis not only synthesizes current knowledge of RACK1 but also pinpoints critical knowledge gaps and promising research directions. Despite significant progress, fundamental questions remain. A pressing goal is to apply in-cell structural biology methods, such as cryo-electron tomography, to visualize specific RACK1 hubs in their native environment and understand how phosphorylation, dimerization, and partner interactions alter its conformation in real time.

For many of RACK1’s post-translational modifications, the responsible kinases and phosphatases in physiological contexts remain unknown. Targeted screening is therefore required to establish a complete “regulatory map,” particularly in neurons and other specialized cells. A central mystery is how RACK1 selectively recruits specific mRNAs for translation within precise cellular compartments in response to specific signals. Addressing this requires new methods to track and manipulate distinct RACK1 pools—ribosomal, membrane, and nuclear—without disrupting cellular homeostasis.

Furthermore, while RACK1 dysfunction is clearly associated with cancer, neurodegenerative, and immune diseases, we must determine whether these disruptions are a primary cause of pathogenesis or a secondary adaptation. It is also critical to clarify how specific mutations or expression changes disrupt the balance between its functional states.

A major methodological challenge is overcoming compartmentalized research approaches. As a quintessential multi-tasker influencing numerous processes, RACK1 cannot be understood in isolation. The field must now transition from quantity to quality, employing systemic analyses that consider context and test functional relationships dynamically. This entails consistently accounting for its ribosomal association, searching for dimeric forms, and developing reporter systems to visualize specific RACK1 states.

A promising direction is the development of therapeutic strategies targeting specific RACK1 functions. The successful identification of an antibody that selectively suppresses cancer cell proliferation demonstrates this approach’s fundamental feasibility. Realizing its full potential, however, requires a detailed functional model to guide the deliberate and controlled discovery of such interventions.

In conclusion, RACK1 emerges not merely as an adapter protein, but as a central regulatory element-a “resource manager” operating at the nexus of translation, signaling, and quality control. Future progress depends on interdisciplinary efforts combining structural biology, biochemistry, cell biology, and genomics. Comprehensive research is now needed to illuminate the full scope of RACK1’s pathways and chart their precise molecular routes, opening new horizons in fundamental cell biology and applied biomedicine.

## Figures and Tables

**Figure 1 ijms-26-11733-f001:**
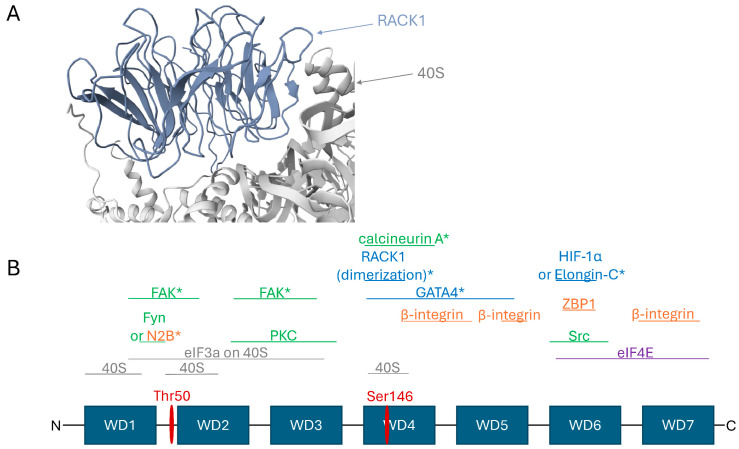
RACK1 can potentially recruit multiple partner proteins while ribosome-bound, though not all simultaneously. (**A**) Structure of RACK1 bound to the human 40S ribosomal subunit (PDB: 4UG0) [127]. RACK1 is positioned on the solvent-exposed surface of the 40S subunit. While a significant portion of its surface mediates ribosome contacts, the remaining area remains accessible for other interactions. (**B**) A linear schematic of RACK1 depicting its domain architecture and key binding sites for selected partners in the ribosome-bound state, based primarily on data from [3] and supplemented as described in the main text. The WD repeats constitute the protein’s structural domains. Regions of RACK1 that are largely occluded by the 40S subunit are indicated; other regions also contact the ribosome but possess substantial exposed surface area facing the cytosol. eIF3a, as a component of the eIF3 complex, binds the 40S subunit and helps organize the surface of initiation complexes. Proteins marked with an asterisk (*) have binding sites that potentially overlap with the ribosome or initiation factors, making their interaction with ribosome-bound RACK1 unlikely. The conjunction “or” denotes mutually exclusive binding partners. Key phosphorylation sites discussed in the text are marked with red lines: phosphorylation of Thr50 promotes RACK1 dissociation from the ribosome, while phosphorylation of Ser146 induces its dimerization. Partner proteins are color-coded: 40S subunit and eIF3a interactors (grey), kinases and phosphatases (green), translation-associated partners (purple), membrane and cytoskeleton-associated partners (orange), and partners involved in regulating gene expression at the transcriptional and translational levels (blue).

**Figure 2 ijms-26-11733-f002:**
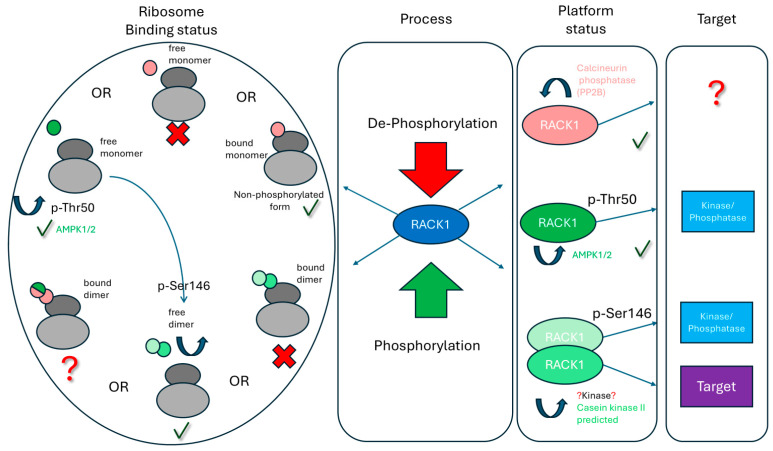
An integrated model of RACK1 function and regulation. Phosphorylated forms of RACK1 are indicated in green; unmodified or dephosphorylated forms are shown in red. Crosses (X) denote unlikely or disfavored states, while question marks (?) indicate transitions or states requiring further validation. RACK1 integrates multiple signaling inputs by concurrently associating with phosphorylating/dephosphorylating enzymes and their target substrates. The dynamic balance between RACK1 functional states is regulated by phosphorylation/dephosphorylation and cellular context, thereby determining phenotypic outcomes. Key transitions include: (1) Monomeric unphosphorylated RACK1 associating with the ribosome; (2) Thr50-phosphorylated RACK1 dissociating from the ribosome; (3) Free RACK1 undergoing dimerization; (4) Ser146-phosphorylated RACK1 being dephosphorylated by calcineurin, restoring monomeric state and ribosomal association to resume translational functions. See main text for detailed explanations.

## Data Availability

No new data were created or analyzed in this study. Data sharing is not applicable to this article.

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
