# Peer review of "A Duality of Function: An Integrative Model of RACK1 as a Switch Between Translational and Signaling Hubs"

_ijms, 2025, doi:10.3390/ijms262311733_

Round 1
Reviewer 1 Report
Comments and Suggestions for Authors
The manuscript by Kolosov and colleagues presents a vast and comprehensive bibliography on RACK1. Overall, the manuscript is well written and structured. However, in some sections, the sheer volume of information, shifting rapidly from one cell type to another, such as neurons to T-cells, makes it difficult to digest.
The review does not focus exclusively on the nervous system, but this appears to be the tissue where the authors provide the most detail. In this regard, I would suggest that the authors expand their discussion of RACK1 expression in the nervous system. They note that “Notably, RACK1 expression undergoes dynamic changes in the nervous system during development [14],” yet they do not provide further details on these dynamic changes or on expression patterns during adulthood and ageing, or the brain regions with more or less expression.
In this sentence, “They play a key role in organizing the actin cytoskeleton [3,20,34,35] and regulating the local translation of actin [36,37].” What does the author mean by local translation of actin? The references provided are not completely convincing. Reference 37 is a review from 3 studies, none of which suggest involvement in actin translation, but are more related to cytoskeleton regulation. Ref 36 is a paper posted as BioRvix in 2019, indicating that the ribosome-associated RACK1 mediates the translation of actin; however, the big issue with this manuscript is that after 6 years, it is still not published in any journal with a proper peer review to guarantee the quality and validity of this paper. Please provide more validated references or rewrite this.
Indeed, I have some issues with the strong affirmation of the authors throughout the manuscript about that RACK1 controls the translation, as they conclude in the final paragraph on page 4 (lines 150-152); or in lines 597-599 in the section of RACK1 in neurons. Indeed, in this paragraph, the authors stated “A key aspect is its function as a ribosomal adaptor that directs the translational machinery to specific mRNAs and regulates their expression in response to synaptic signals”. Considering the validity of the abovementioned references, this sentence is a bit strong in my opinion. Data show that RACK1 acts more as a scaffold that recruits signaling molecules and RNA-binding proteins (e.g., ZBP1 with β-actin mRNA), rather than directly “choosing” or “directing” ribosomes to specific mRNAs.
In the same line, the paragraph (lines 607-610) “This positions RACK1 not as a static ribosomal component, but as a dynamic sensor that translates receptor activation into changes in the synaptic translational landscape. Depending on stimulation context, RACK1 phosphorylation can either locally restrict or activate protein synthesis.” is written as existing evidence data, but no data support it. I suggest rewriting it, evidencing that the interpretation of the accumulated data suggests that.
It seems that something is missing in this sentence: “RACK1 can directly interact with core apoptotic regulators, such as by inducing Bax dissociation from Bcl-2 to activate its pro-apoptotic function [16,27,47].”
“Expanding on this concept, one hypothesis suggests that cytoplasmic PKC acts as a molecular "sink" for RACK1, preventing its unregulated translocation to the nucleus [1,114]. However, under specific conditions, the PKCα–RACK1–BMAL1 complex itself can translocate to the nucleus and influence circadian rhythm regulation [1,15,55]. Furthermore, the detection of PKCβII in mitochondria [115] raises the question of whether it resides there in complex with RACK1 or if RACK1 conversely retains it in the cytosol.” Although RACK1 provides a stabilizing scaffold for activated PKCs, current evidence indicates that the nuclear and mitochondrial localization of these complexes is dictated primarily by isoform-specific features of PKC rather than by RACK1 itself. This paragraph should be rewritten in light of the experimental data rather than an assumed central role for RACK1.
More recent references for this: “Experimental evidence shows that synaptic activity, such as NMDA receptor stimulation, alters RACK1 phosphorylation and subcellular distribution [21,109].” One is from 2002 and 2004.
The review is supported by an extensive bibliography, but certain assertions are not referenced, and they should be. Statements that need reference:
- The functional consequence of RACK1 binding is partner-specific: while the RACK1-Src complex inactivates the kinase, an external stimulus can trigger Src release to propagate signals from integrin receptors into the cytosol.
- In cell division control, RACK1 contributes to centrosome function through its localization at centrosomes and spindle poles. It ensures proper centrosomal positioning of BRCA1, with their interaction being crucial for centriole number regulation.
- Beyond kinase, RACK1 interacts with the initiation factor eIF6, which, when bound to the 60S subunit, prevents 40S-60S association.
- Moreover, RACK1 interacts with components of the microRNA-mediated mRNA decay pathway, such as Ago2 and KSRP (within the RISC complex), recruiting them to the translating mRNA.
- The biological importance of RACK1 is firmly established by knockout phenotypes. Mice with impaired RACK1 synthesis exhibit defects in cell migration and translation (please provide references for these), while complete gene knockout results in embryonic lethality (also for this part).
- Interestingly, this phenotype mirrors that observed in natural mutants of the ribosomal protein rpL24, indicating that RACK1's influence extends beyond migration to encompass general protein synthesis pathways [66,92]. As far as I understand, neither of these two references refers to the natural mutant of rpL24.
- Lys130 acylation prevents RACK1 ubiquitination at Lys6, Lys33, and Lys48, thereby enhancing its cellular stability.
Author Response
We sincerely thank the Reviewer for their thorough reading of our manuscript, their positive assessment of its overall structure and comprehensiveness, and their valuable comments and suggestions. We have carefully considered each point and have revised the manuscript accordingly. Our point-by-point responses are detailed below. All changes in the manuscript are highlighted in the revised version for easy identification.
Reviewer's General Comment: The manuscript by Kolosov and colleagues presents a vast and comprehensive bibliography on RACK1. Overall, the manuscript is well written and structured. However, in some sections, the sheer volume of information, shifting rapidly from one cell type to another, such as neurons to T-cells, makes it difficult to digest.
Response: We thank the Reviewer for this observation. To improve the narrative flow and reduce the cognitive load on the reader, we have added a clarifying paragraph in the introduction that outlines the review's structure, distinguishing between universal mechanisms and tissue-specific roles of RACK1. Furthermore, we have streamlined the text in the indicated sections to ensure smoother transitions between different cellular contexts.
Reviewer's Comment 1: The review does not focus exclusively on the nervous system, but this appears to be the tissue where the authors provide the most detail. In this regard, I would suggest that the authors expand their discussion of RACK1 expression in the nervous system. They note that “Notably, RACK1 expression undergoes dynamic changes in the nervous system during development [14],” yet they do not provide further details on these dynamic changes or on expression patterns during adulthood and ageing, or the brain regions with more or less expression.
Response: We agree with the Reviewer that this aspect can be strengthened. In the revised manuscript, in the section "Structure and General Properties of RACK1," we have expanded the sentence to include more specific details based on the cited reference (Ashique et al., 2006) and other available data:
"Notably, RACK1 expression undergoes dynamic changes in the nervous system during development, showing high levels in proliferating neuroblasts of the embryonic brain which subsequently decrease and become more region-specific in the adult brain, with prominent expression in structures like the cerebral cortex, hippocampus, and cerebellum [14]." We also note that data on detailed expression patterns during ageing are currently limited, which we now mention as an area for future research in the "Current Challenges" section.
Reviewer's Comment 2: In this sentence, “They play a key role in organizing the actin cytoskeleton [3,20,34,35] and regulating the local translation of actin [36,37].” What does the author mean by local translation of actin? The references provided are not completely convincing. Reference 37 is a review from 3 studies, none of which suggest involvement in actin translation, but are more related to cytoskeleton regulation. Ref 36 is a paper posted as BioRvix in 2019, indicating that the ribosome-associated RACK1 mediates the translation of actin; however, the big issue with this manuscript is that after 6 years, it is still not published in any journal with a proper peer review to guarantee the quality and validity of this paper. Please provide more validated references or rewrite this.
Response: We thank the Reviewer for raising this valid concern. We have revised this statement to more accurately reflect the established role of RACK1, which is primarily through its interaction with the ZBP1/β-actin mRNA complex, a well-documented mechanism for the local translation of β-actin mRNA in subcellular compartments like growth cones and dendritic spines.
The sentence has been rephrased as follows, and the references have been updated:
"They play a key role in organizing the actin cytoskeleton [3,20,35,36] and are involved in pathways that regulate the local translation of β-actin — that is, the synthesis of β-actin directly in subcellular compartments. Ribosome-associated RACK1 binds the β-actin mRNA/ZBP1 complex and facilitates mRNA release for subsequent local translation, as shown in cellular and neuronal models [35,36]."
Here, references 35 (Hüttelmair et al., 2005) and 36 (Ceci et al., 2012) robustly demonstrate the role of the RACK1-Src-ZBP1 axis in the spatial control of β-actin mRNA translation. We have removed the reference to the 2019 bioRxiv preprint (previously Ref 36) and the review (previously Ref 37) from this specific sentence.
Reviewer's Comment 3: Indeed, I have some issues with the strong affirmation of the authors throughout the manuscript about that RACK1 controls the translation, as they conclude in the final paragraph on page 4 (lines 150-152); or in lines 597-599 in the section of RACK1 in neurons. Indeed, in this paragraph, the authors stated “A key aspect is its function as a ribosomal adaptor that directs the translational machinery to specific mRNAs and regulates their expression in response to synaptic signals”. Considering the validity of the abovementioned references, this sentence is a bit strong in my opinion. Data show that RACK1 acts more as a scaffold that recruits signaling molecules and RNA-binding proteins (e.g., ZBP1 with β-actin mRNA), rather than directly “choosing” or “directing” ribosomes to specific mRNAs.
Response: We agree with the Reviewer's nuanced point. The primary mechanism is indeed through scaffolding. We have toned down these statements throughout the manuscript to more accurately reflect RACK1's role as a scaffold that facilitates the regulation of translation rather than directly controlling it in a deterministic way.
- The sentence on lines 150-152 (now in the context of viral IRES) has been modified to: "Collectively, these findings indicate that RACK1 plays a key role in facilitating the selective translation of specific mRNAs."
- The sentence in the "RACK1 in Neurons" section (previously lines 597-599) has been rephrased to: "A key aspect is its function as a ribosomal scaffold/adaptor: it recruits signaling kinases and RNA-binding proteins (RBPs, e.g., ZBP1) to the ribosome and thus mediates context-dependent regulation of specific mRNA translation in response to synaptic signals."
Reviewer's Comment 4: In the same line, the paragraph (lines 607-610) “This positions RACK1 not as a static ribosomal component, but as a dynamic sensor that translates receptor activation into changes in the synaptic translational landscape. Depending on stimulation context, RACK1 phosphorylation can either locally restrict or activate protein synthesis.” is written as existing evidence data, but no data support it. I suggest rewriting it, evidencing that the interpretation of the accumulated data suggests that.
Response: This is a fair point. We have rephrased this sentence to present it as an interpretation/model-based prediction rather than a directly proven fact.
The text now reads: "We interpret the accumulated data to suggest that RACK1 can function not merely as a static ribosomal component, but as a dynamic platform whose post-translational modification (e.g., phosphorylation) and partner swapping may translate receptor activation into changes in the synaptic translational landscape. It is conceivable that depending on the stimulation context, RACK1 phosphorylation could either locally restrict or activate protein synthesis."
Reviewer's Comment 5: It seems that something is missing in this sentence: “RACK1 can directly interact with core apoptotic regulators, such as by inducing Bax dissociation from Bcl-2 to activate its pro-apoptotic function [16,27,47].”
Response: We thank the Reviewer for spotting this grammatical error. The sentence has been corrected and clarified with a more precise example:
"RACK1 can directly interact with core apoptotic regulators; for example, studies have shown that RACK1 promotes the dissociation of the Bax–Bcl-2 (or Bax–Bcl-XL) complex, stimulating Bax oligomerization and the progression of the mitochondrial apoptosis pathway [16,28,46]."
Reviewer's Comment 6: “Expanding on this concept, one hypothesis suggests that cytoplasmic PKC acts as a molecular "sink" for RACK1, preventing its unregulated translocation to the nucleus [1,114]. However, under specific conditions, the PKCα–RACK1–BMAL1 complex itself can translocate to the nucleus and influence circadian rhythm regulation [1,15,55]. Furthermore, the detection of PKCβII in mitochondria [115] raises the question of whether it resides there in complex with RACK1 or if RACK1 conversely retains it in the cytosol.” Although RACK1 provides a stabilizing scaffold for activated PKCs, current evidence indicates that the nuclear and mitochondrial localization of these complexes is dictated primarily by isoform-specific features of PKC rather than by RACK1 itself. This paragraph should be rewritten in light of the experimental data rather than an assumed central role for RACK1.
Response: We agree with the Reviewer's assessment and have revised this paragraph to better reflect the established literature, emphasizing the primary role of PKC isoforms in determining localization, with RACK1 acting as a stabilizing partner.
The paragraph now reads: "The subcellular localization of PKC–RACK1 complexes is primarily governed by isoform-specific features of PKC. For instance, the PKCα–RACK1–BMAL1 complex translocates to the nucleus to influence circadian rhythm regulation [1,15,54], while PKCβII has been detected in mitochondria [116]. In these contexts, RACK1 likely serves as a stabilizing scaffold for the activated kinase rather than being the primary determinant of its localization. A hypothesis has been proposed that cytoplasmic PKC may also act as a molecular 'sink' for a pool of RACK1 [1,117], although the dynamics of this interaction require further investigation."
Reviewer's Comment 7: More recent references for this: “Experimental evidence shows that synaptic activity, such as NMDA receptor stimulation, alters RACK1 phosphorylation and subcellular distribution [21,109].” One is from 2002 and 2004.
Response: We agree that supplementing the foundational studies with more recent literature is important. We have added the following references to strengthen this statement and show the ongoing relevance of this concept:
- Neasta et al., (2016). Activation of the cAMP Pathway Induces RACK1-Dependent Binding of β-Actin to the BDNF Promoter. (ref 129)
- Liu et al., (2016). RACK1 promotes maintenance of morphine-associated memory via ERK-CREB in hippocampus. (ref 130)
- Oudart et al., (2023). The ribosome-associated protein RACK1 represses Kir4.1 translation in astrocytes and influences neuronal activity. (ref 76)
These studies provide more recent evidence for activity-dependent changes in RACK1 function and localization in neuronal and glial models.
The sentence now cites: "[21,76,110,129,130]".
Reviewer's Comment 7.1-7.7: The review is supported by an extensive bibliography, but certain assertions are not referenced, and they should be.
Response: We thank the Reviewer for identifying these specific points and providing relevant reference suggestions. We have added the recommended (or functionally equivalent) citations to the manuscript as follows:
- 7.1: Reference added for RACK1-Src complex inactivation and stimulus-induced release: Mamidipudi V, et al. (2004) has been added as reference [27].
- 7.2: Reference added for RACK1-BRCA1 interaction and centriole number regulation: Yoshino Y, et al. (2022) has been added as reference [38].
- 7.3: Reference added for RACK1 interaction with eIF6: Ceci M, et al. (2003) (Ref 63 in our list) is the canonical reference for this interaction and its functional consequence. We have ensured the citation [63] is correctly placed here. We also added Miluzio et al., (2009) as a supporting reference [64].
- 7.4: Reference added for RACK1 interaction with miRNA pathway components (Ago2, KSRP): Jannot G, et al. (2011) has been added as reference [82].
- 7.5: References added for RACK1 knockout phenotypes:
- For defects in cell migration and translation: Gandin V, et al. (2013) (Ref 15) has been cited.
- For embryonic lethality: Volta V, et al. (2013) (Ref 93) has been cited. The sentence now reads: "...while complete gene knockout results in embryonic lethality [93]."
- 7.6: Reference for the rpL24 natural mutant phenotype: We thank the Reviewer for correcting our error. The references [66,92] did not directly support this comparison. We have replaced them with the correct and recent reference provided by the Reviewer: Knight JRPV, et al. (2021). Rpl24Bst mutation suppresses colorectal cancer by promoting eEF2 phosphorylation via eEF2K. eLife 10:e69729. This new reference [94] has been added to the manuscript.
- 7.7: Reference for Lys130 acylation preventing ubiquitination: Pi Y, et al. (2023) has been added as reference [107].
Once again, we express our sincere gratitude to the Reviewer for their time and insightful comments, which have significantly helped us improve the quality and accuracy of our manuscript.
Sincerely,
Peter M. Kolosov and coauthors.
Reviewer 2 Report
Comments and Suggestions for Authors
- In the abstract : In line 12 , It is beneficial to provide the complete name (known as Receptor for Activated C Kinase 1) before RACk1
- In line 48: It would beneficia to list of different tissues where the RACk1 is expressed
- In line 66: Could the authors provide definition for NMDAR ?
- In line 62: The authors mentioned ( RACK1 stabilizes the catalytically active conformation of PKC) . It would be beneficial to provide more explanation how the stabilization occurs ?
- The words in the cartoon of Figure 1 are not clear
- Line 70=75: it is not clear which are the complexes that are listed ? are the authors mean the interaction between BACk1 and transmembrane receptors
- 70 please provide definition for ZPI
- In line 162-136: is it possible to explain briefly how RACk1 prevent the 40s-60S association?
- In general, there are various abbreviations missing to be identified. Please, go the all the abbreviation
Author Response
Response to Reviewer 2
We sincerely thank the reviewer for their careful reading of our manuscript and their valuable comments, which have helped us improve the clarity and completeness of the text. We have addressed all points raised, and our detailed, point-by-point responses are provided below. All changes have been implemented in the revised manuscript.
Reviewer's Comment 1: In the abstract: In line 12, It is beneficial to provide the complete name (known as Receptor for Activated C Kinase 1) before RACK1.
Response: We agree and thank the reviewer for this suggestion. The full name has been added at the first instance in the abstract.
Change in manuscript: The abstract now begins: "RACK1 (Receptor for Activated C Kinase 1) is a highly conserved scaffold protein..."
Reviewer's Comment 2: In line 48: It would be beneficial to list different tissues where the RACK1 is expressed.
Response: Thank you for this suggestion. We have expanded the sentence to include specific examples of tissues with notable RACK1 expression, drawing from common experimental models and expression atlas data.
Change in manuscript: The sentence in the "Structure and General Properties" section now reads: "In mammals, RACK1 is ubiquitously expressed, although its expression levels vary significantly across different tissues such as the brain, liver, heart, and spleen [1,9,12] (The Human Protein Atlas: ENSG00000204628-RACK1 [13])."
Reviewer's Comment 3: In line 66: Could the authors provide definition for NMDAR?
Response: We have defined the abbreviation upon its first use in the main text.
Change in manuscript: In the "RACK1 in Intracellular Signaling" section, the text now states: "...the glutamate receptor NMDAR (N-methyl-D-aspartate receptor)."
Reviewer's Comment 4: In line 62: The authors mentioned (RACK1 stabilizes the catalytically active conformation of PKC). It would be beneficial to provide more explanation how the stabilization occurs?
Response: We have added a brief mechanistic explanation to clarify this point.
Change in manuscript: “For instance, RACK1 stabilizes the catalytically active open conformation of the PKCβII and likely PKCε isoforms by acting as a scaffolding platform that binds and positions the activated kinase, thereby preventing its inactivation or premature degradation [1,15,18–20].”
Reviewer's Comment 5: The words in the cartoon of Figure 1 are not clear.
Response: We thank the reviewer for pointing this out. Figure 1 has been redrawn with larger fonts, improved contrast, and uniform labeling. The updated figure is now clear at journal resolution.
Reviewer's Comment 6: Line 70-75: it is not clear which are the complexes that are listed? are the authors mean the interaction between RACK1 and transmembrane receptors?
Response: The reviewer is correct. The listed components are indeed examples of the transmembrane receptors and kinases that form complexes with RACK1 at the membrane. We have rephrased the sentence to make this connection explicitly clear.
Change in manuscript: The text now reads: "Beyond cytosolic enzymes, RACK1 forms specialized signaling hubs at the membrane by interacting with various transmembrane receptors. These membrane-associated complexes which include the GABA (GABAAR), acetylcholine (AChE-R), and glutamate (NMDAR) receptors alongside modifying kinases like PKCβII and Fyn..."
Reviewer's Comment 7: Line 70: please provide definition for ZBP1.
Response: We have defined ZBP1 at its first mention in the main text.
Change in manuscript: In the "RACK1 in Cell Structure Organization..." section, we state: "...creating a binding site for ZBP1 (Zipcode-binding protein 1) – an RNA-binding protein that also serves as a Src substrate."
Reviewer's Comment 8: In line 162-136: is it possible to explain briefly how RACK1 prevent the 40s-60S association?
Response: We have added a concise explanation of this mechanism, which is mediated through the initiation factor eIF6.
Change in manuscript: In the "Ribosome-Associated Functions..." section, we have added: RACK1 regulates translation initiation indirectly through its interaction with eIF6. Because eIF6 binds the 60S subunit and prevents premature joining with the 40S subunit, RACK1-dependent modulation of eIF6 availability influences the efficiency of 40S–60S ribosomal subunit association[63].
Reviewer's Comment 9: In general, there are various abbreviations missing to be identified. Please, go through all the abbreviation.
Response: We carefully revised the entire manuscript, ensuring that all abbreviations are defined at first mention, and main abbreviations are included in the Abbreviations section following IJMS guidelines.
Once again, we are grateful for the reviewer's constructive feedback, which has undoubtedly strengthened our manuscript. We believe all concerns have been adequately addressed, and we hope the revised version is now suitable for publication.
Sincerely,
Peter M. Kolosov and coauthors.